# Non-Pharmacological Treatment for Cardiovascular Risk Prevention in Children and Adolescents with Obesity

**DOI:** 10.3390/nu16152497

**Published:** 2024-07-31

**Authors:** Simonetta Genovesi, Andrea Vania, Margherita Caroli, Antonina Orlando, Giulia Lieti, Gianfranco Parati, Marco Giussani

**Affiliations:** 1School of Medicine and Surgery, University of Milano-Bicocca, 20126 Milano, Italy; parati@auxologico.it; 2Istituto Auxologico Italiano, IRCCS, 20145 Milano, Italy; a.orlando@auxologico.it (A.O.); m.giussani@auxologico.it (M.G.); 3Independent Researcher, 00162 Rome, Italy; andrea.vania57@gmail.com; 4Independent Researcher, Francavilla Fontana, 72021 Brindisi, Italy; margheritacaroli53@gmail.com; 5UO Nefrologia e Dialisi, ASST-Rhodense, 20024 Garbagnate Milanese, Italy; giulialieti@gmail.com

**Keywords:** adolescents, cardiovascular risk factors, children, obesity, prevention

## Abstract

In younger generations, excess weight has reached very alarming levels. Excess weight in adults is associated with increased mortality and morbidity from cardiovascular disease. However, it is not easy to distinguish to what extent these effects are the result of obesity itself or how much is due to the various cardiovascular risk factors that often accompany excess weight. Several risk factors, such as hypertension, dyslipidemia, hyperuricemia, glucose intolerance, and type 2 diabetes mellitus, are already present in pediatric age. Therefore, early intervention with the goal of correcting and/or eliminating them is particularly important. In the child and adolescent with obesity, the first approach to achieve weight reduction and correct the risk factors associated with severe excess weight should always be non-pharmacologic and based on changing poor eating habits and unhealthy lifestyles. The purpose of this review is to give an update on non-pharmacological interventions to be implemented for cardiovascular prevention in children and adolescents with obesity, and their effectiveness. In particular, interventions targeting each individual cardiovascular risk factor will be discussed.

## 1. Introduction

Weight excess is reaching alarming levels amongst the younger generations. In Italy, 9.8% of 8-year-old children are obese and 19% of them are in overweight [1]; in the USA, 19.7% of children and adolescents between 2 and 19 years old of age suffer from obesity [2,3]; and even in countries with a low economic standard, the prevalence of weight excess and obesity in childhood is high [4,5]. If no specific interventions are undertaken at a young age, weight excess tends to persist from childhood into adulthood [6,7]. Obesity facilitates the onset of several cardiovascular risk factors (CVRF) that are frequently observed as early as childhood and adolescence [8]. Globally, cardiovascular diseases (CVD) represent the leading cause of death [9,10] implicating high healthcare costs [11]; when the high number of children that are currently overweight will become adults, these costs will presumably rise to such a level that even the welfare systems of developed countries will not have the necessary funds to sustain the expenses due to expected increase in cardiovascular diseases associated with obesity [10]. In poorer countries, the problem will become even more serious, as their economic resources aimed at curing noncommunicable diseases are already scarce and insufficient [11,12]. Against this background, it is clearly very important to start efficient prevention of CVD as early as childhood and continuing into adolescence. Preventive activity should take place at two distinct levels. On the one hand, it should involve all children and adolescents in campaigns that promote and spread the principles of an appropriate diet and of a healthy lifestyle, in order to counteract the effects of the *Western Diet* and of the sedentary habits that are increasingly frequent in pediatric age [13,14]. On the other hand, at a later stage, when CVRFs have already emerged, it will be necessary to implement individual interventions to identify and correct these risk factors. This second point is particularly important, as many parents fail to recognize their children’s obesity as a disease [15]. By identifying the obesity-related CVRFs in their children, parents and other family members may become more aware of the problem and consider the children’s weight excess in a more objective and critical manner. In this review, while being aware of the importance of several other diseases that may be related to obesity [16,17], we will focus on weight excess-related CVRFs and on possible early cardiovascular disorders. We will also discuss the targeted interventions to correct these clinical conditions, which are mainly based on dietary and behavioural changes and, only rarely, on pharmacological treatment. The basic concept of these interventions would be to shift the focus of obesity treatment from simply losing weight towards a more global approach that aims at reducing, and possible eliminating, cardiovascular risk. From this perspective, weight loss represents only one of the treatment goals.

## 2. Obesity as a Cardiovascular Risk Factor

Weight excess has been shown to be related to a conspicuous increase in mortality and morbidity from cardiovascular causes [18]. It is not easy, however, to distinguish to what degree this effect is due to obesity per se or to the presence of different CVRFs that are often associated with excess weight (hypertension, dyslipidemia, hyperuricemia, glucose intolerance, and diabetes mellitus [T2DM]). In other words, is obesity a CVRF in itself, or is it rather an activator of more dangerous risk factors? To answer this question, it is necessary to make the following considerations. First of all, it is fundamental to know if subjects with obesity and without other CVRFs show signs of cardiovascular organ damage and if the onset of this damage occurs as early as in childhood or during adolescence. Furthermore, it is important to understand if the incidence of cardiovascular events is different in metabolically healthy individuals affected by obesity (Metabolically Healthy Obesity [MHO], i.e., without any sign of metabolic alterations that define metabolic syndrome [MS]) compared with normal-weight persons. The definition of MHO, which is widely used in adults with severe excess weight [19], has also been proposed for children and adolescents [20]. However, in our opinion, the distinction between an MHO and a metabolically unhealthy (MUO) phenotype can be ambiguous and inaccurate when used for pediatric populations. While this distinction can potentially be useful in identifying children and adolescents most at risk, it may lead to underestimating the number of children in whom obesity per se has led to early organ damage even in the absence of metabolic alterations or is associated with the presence of metabolic risk factors not included in the definition, such as hyperuricemia and/or insulin resistance.

A study that involved 459 children affected by obesity with an average age of 10.6 ± 2.6 years, revealed the presence of left ventricular hypertrophy (LVH) in a high percentage of the study population. The prevalence of LVH in children and adolescents with MHO was slightly lower (31.1 vs. 40%, *p* = 0.06) than that observed in peers with MUO and the only factors that were shown to be significantly associated with LVH prevalence were body mass index (BMI) z-score and Waist-to-Height-ratio. The authors concluded that the MUO phenotype could not be considered a useful indicator of the presence of early organ damage in their study population [21]. The same research team subsequently demonstrated that in pediatric subjects the relationship between weight excess and LVH is stronger than the association between hypertension (HT) and LVH [22,23]. Children and adolescents with MHO present with a more unfavorable metabolic profile than their normal weight peers [24]; moreover, hyperinsulinemia and hyperuricemia are frequently observed in MHO [25]. Lastly, a large study in children and adolescents of 6 to 17 years old proved that the carotid intima-media thickness was higher in subjects with MHO compared with contemporaries with normal weight [26]. These results suggest that pediatric age obesity has a negative influence on metabolic parameters and on markers of cardiovascular organ damage.

If, however, we want to investigate if MHO is related to higher incidences of cardiovascular mortality and morbidity compared with the normal-weight state, we must refer to the observations made in adult populations. Several studies performed in large populations [27,28,29,30,31] and some meta-analyses [32,33] confirm that persons with MHO have a higher risk of developing cardiovascular disease (CVD) and a higher mortality rate compared with metabolically healthy individuals without excess weight. Some studies, carried out on smaller study populations and with shorter follow-up durations, showed that these differences were evident only among male subjects [34] and one other study failed to demonstrate a higher incidence of cardiovascular events in persons with MHO compared with normal-weight individuals without any risk factors. In any case, in this study, the metabolomic profiles of the subjects with MHO suggested an increased cardiovascular risk [35].

## 3. The Non-Pharmacological Treatment Approach in Children and Adolescents with Obesity

From what was described in the previous section, it appears clear that, while it is well known that in subjects with obesity the presence of other CVRF increases the risk of cardiovascular events, obesity per se needs to be considered an independent CVRF with very precocious negative effects. For this reason, it is not a surprise to observe that there are different opinions regarding the distinction between MHO vs. MUO. The American Academy of Pediatrics (AAP) does not mention these definitions in its recent Guidelines [16] and considers all children and adolescents affected by obesity at equal risk of developing metabolic complications and early cardiovascular damage. The AAP acknowledges that obesity is a chronic disease with negative effects that advance and accumulate in time, emphasizes the importance of thorough examinations and evidence-based interventions, and affirms that children with obesity or overweight strongly benefit from assuming a healthy lifestyle and behaviour. This “life course approach” should focus not only on the children but also on their families and should start as soon as possible and be continued throughout childhood and adolescence until the adult age. The point of view of the AAP is not very different from the 2015 position of the EASO (European Association for the Study of Obesity) [36]. The EASO also underlines the importance of a therapeutic approach to obesity which, independently of its severity and of the presence or not of complications and comorbidities, should always include and start with changes in lifestyle and nutrition. Both scientific societies also agree [16,36] on the fact that, in selected cases, the use of drug therapy or even bariatric surgery may be considered, similarly to what is done in adult patients [37]. The application of these two additional steps depends on the severity of the obesity and on the presence of concomitant diseases and represents an issue that goes beyond the scope of the current review.

When addressing the topic of lifestyle changes, social differences and inequities (defined as “the lack of economic, civil-political, cultural, or environmental conditions that are required to generate parity and equality” [38] that cause them cannot be ignored. As a matter of fact, social injustices experienced throughout life may have a negative impact on both the genesis, persistence, and worsening of obesity [16]. Over the years, the issue of the influence of social disparities and injustices on pediatric obesity has been the subject of many studies [39,40,41,42], sometimes with unexpected results, such as the observation that being bullied may induce chronic low-grade systemic inflammation [43], a condition that is considered to be a CVRF [44].

As described earlier, the first step in preventing and treating obesity should be to achieve healthier eating habits and lifestyle. When we look at prevention, the creation of good feeding practices involves the entire family and should start from early childhood, by applying “responsive feeding” practices, such as indicated by the WHO [45], beginning with proper breast-feeding routines and continuing during complementary feeding and at all later stages of development. It is important to remember that many characteristics of complementary foods may have an impact on the children’s health both in the short and in the long term [46]. For this reason, it is of fundamental importance to establish right away what is correct feeding in terms of quantity and quality of food, and also in terms of the manner in which food is administered. For nutrition of children after the stage of complementary feeding, the “Mediterranean diet” [47], inserted in 2010 in the UNESCO List of the Intangible Cultural Heritage of Humanity [48], is currently considered the healthiest dietary pattern regarding overall well-being and prevention from CVRF.

The Mediterranean diet is acknowledged for its influence on general lifestyle, promoting conviviality, intercultural dialogue, respect for territorial biodiversity, and cultural traditions. As UNICEF highlights in another document [49], the Mediterranean diet is much more than a list of foods or a nutritional chart. It is a lifestyle. The Mediterranean diet contrasts with less healthy eating styles, such as the “Western diet” [50], which is rich in saturated fats, refined carbohydrates, ultra-processed foods, and salt, and increasingly prevalent, especially in industrialized countries. The “Western diet” has been associated with an increased prevalence of metabolic disorders such as obesity, diabetes, and CVD, as well as other pathologies associated with or conditioning the former, such as cognitive impairment, emotional disorders, depression, anxiety, and chronic stress [51,52].

Regarding nutritional therapy of pediatric obesity, several national [53,54,55,56,57] and international [58,59,60] guidelines emphasize that establishing healthy lifestyle and eating behaviours is a priority over merely losing excess weight. It is also underlined that nutritional intervention should be adjusted according to the degree of excess weight (overweight or first-degree obesity vs. higher-degree obesity), the age of the individual, and the presence or absence of complications. The goal of the nutritional intervention should be to reduce overall energy intake and to help, through food education that involves the entire family, increase the consumption of healthy foods, such as vegetables, legumes, and fruits, thus helping to reduce the consumption of ultra-processed foods and foods that are too rich in fats and simple sugars. It is important to note that very restrictive diets are contraindicated.

The education process towards better lifestyle and eating habits may be facilitated by psychological support, ranging from psychological counseling to psychotherapy; such support may be offered to the individual or, when necessary and appropriate, to the entire family [16,58,59,61,62], although not all authors agree on the latter option [63]. Behavioural psychosocial counseling supports the education process for healthier lifestyles and offers mechanisms to counterbalance any negative effects of the changes that are being proposed during the process. Although it is always desirable to have a psychologist in the team dealing with pediatric obesity [16,58,59], the ability to address and manage these issues should be an integral part of the training and skills of the pediatricians within this team.

### 3.1. Obesity and Hypertension

Several epidemiological studies have shown that elevated blood pressure values are not rare in school-age children and in adolescents. While studies that have evaluated the prevalence of hypertension (HT) in this age range differ from a methodological point of view and comparisons cannot easily be made, it can nevertheless be affirmed that HT prevalence in the pediatric age amounts to about 4.0% [64]; it should be noted that this value is strongly influenced by the prevalence of overweight. Screening programs in schools with repeated and controlled blood pressure measurements report that among children with normal weight, hypertension prevalence was somewhat higher than 1%, arriving at 4–5% in overweight peers and reaching about 20% in children with obesity, with only negligible differences between boys and girls [64,65,66]. A recent meta-analysis [67] performed on 47 studies confirms these data, showing an overall prevalence of hypertension of 4.0% (1.9% in normal weight, 5.0% in overweight, and 15.3% in obesity), without any differences between gender and city versus rural living environment. The meta-analysis reveals a significant increase in the HT prevalence from the 1990s, probably in association with the marked growth in the frequency of excess weight among children and adolescents. It has been shown that if high blood pressure values observed in pediatric age are not treated, they will tend to remain high in adulthood [68,69]. The pathophysiology of essential hypertension is complex and not yet completely clear. We do know, however, that in overweight children, besides daily intake of sodium and simple sugars [70], levels of insulin resistance and uric acid may play an essential role [71,72]. These observations have an important impact on how correct dietary treatment approaches are defined. For overweight children and adolescents, just as for adults, the AAP [16] and the National Heart, Lung, and Blood Institute in the USA [73] advise to follow the DASH (Dietary Approaches to Stop Hypertension) recommendations [74,75] both for prevention in individuals with a positive family history for hypertension and for treatment. DASH suggests a physical activity program for the prevention and cure of HT; returning to normal weight in cases of overweight and reducing the consumption of foods with high saturated fats, hydrogenated fats, and cholesterol as well as lowering glucose loads. Furthermore, DASH recommends the promotion of the consumption of foods that have high potassium and magnesium contents, high levels of essential ω3 fatty acids, and low levels of sodium, as well as a reduction of the use of kitchen salt (sodium chloride) and foods in which salt is used as a preservative. The elimination of tobacco and alcohol consumption is also recommended by DASH; this is especially important for adolescents, who display a high frequency of tobacco and alcohol use [76,77,78]. Regarding sodium intake in particular, the guidelines for the treatment of pediatric hypertension suggests limiting the consumption of sodium to less than 2 g/die (5 g of salt) by reducing the amount of salt already present in foods [79,80].

### 3.2. Obesity and Dyslipidaemia

Dyslipidaemia, if searched for, is not rare in pediatric age. Among North American children aged 8–17 years, 20.2% have a form of dyslipidaemia. The prevalence increases from 14.6% among normal-weight children to 39.3% among those with obesity [81]. A large European study confirms these data, placing the prevalence of dyslipidaemia in subjects with obesity between 38.2% in females and 40.5% in males in the age group of 6–14 years [82]. In Italy too, cholesterol and triglyceride levels are higher among children with overweight or obesity [83].

As in adults, the most frequent form of dyslipidaemia in children with obesity is a decrease in HDL cholesterol, usually associated with an increase in triglyceride levels, both alterations that are typical of metabolic syndrome (MS); this form of dyslipidaemia is also known as atherogenic dyslipidaemia because it has been shown that the combination of these two lipid alterations, potentiated by the presence of small dense LDL, promotes atherosclerosis [84]. Isolated hypercholesterolaemia is also common in overweight children and is involved in atherosclerotic processes as well. In the vast majority of cases, isolated hypercholesterolaemia can be caused by two conditions: familial hypercholesterolemia or the polygenic form. Familial hypercholesterolemia is not influenced by weight class, as it has an exclusively genetic aetiology. In contrast, the more common polygenic hypercholesterolemia requires that incorrect lifestyle and dietary habits be added to a genetic predisposition, and for this reason, the condition develops predominantly in overweight children [85]. As suggested by American guidelines [86], the identification and subsequent treatment of children with dyslipidaemia is important because it has been shown that, already in pediatric age, these lipid alterations favor the thickening of the carotid intima-media, which tends to persist into adulthood even if the lipid profile normalizes [87].

Studies agree that both polygenic forms of dyslipidaemia and heterozygous forms of familial hypercholesterolaemia can greatly benefit from nutritional approaches, regarding both prevention and therapy [88,89,90,91,92]. Such an approach should therefore represent the first step in treating these disorders, especially when they are accompanied by overweight or a concomitant metabolic syndrome. The first intervention in these nutritional approaches, emphasizing to patients and families the importance of adhering to a “healthy diet”, particularly the Mediterranean diet, the Nordic Diet (with nutrition advice adapted to local intake patterns [93]), or the DASH model [74], may already be sufficient to improve the lipid profile. In fact, a “healthy diet” limits the intake of saturated fats, trans fats, and cholesterol [73,89]. Essentially, a “healthy diet” aimed at preventing or treating dyslipidaemia involves (a) a varied diet rich in fruits (but not in the form of juices, blended, or freshly pressed fruits), vegetables, and unrefined cereals, with an adequate or moderate intake of proteins from lean meat, fish, eggs, legumes, certain types of oil seeds, and also low-fat dairy products; (b) a reduced intake of saturated fats, such as those from red or fatty meats, high-fat dairy products, cooking fats other than olive oil, and avoiding trans fats, often found in ultra-processed foods; and (c) the exclusion of sugar-sweetened beverages such as soda and sports drinks, as well as sweetened teas or herbal infusions.

It is important to realize that even apparently healthy beverages like fruit juices and blended or freshly pressed fruits can be a source of simple sugars [88,89]. It should be noted that more mild dietary approaches, such as the “traffic light” scheme [89] that distinguishes between foods to prefer (green light), foods to use in moderation (yellow light), and foods to consume only occasionally (red light), can be useful and highly tolerable, for example, in cases of mild lipid alterations in children and adolescents with overweight or grade I obesity. If initial dietary changes prove ineffective, adhering to more structured and restrictive dietary protocols such as the CHILD-1 or its upgrades CHILD-2-LDL-C 72 and/or CHILD-2-TG diet [90], with further restrictions of saturated fats and cholesterol and emphasis on increased fiber intake, can have additional benefits on the serum lipid profile.

In children or adolescents with obesity and dyslipidaemia, the failure of dietary regimens, often due to poor adherence, does not necessarily entail the need for an immediate shift to drug therapy. There are several nutraceuticals and other non-pharmacological substances that can be used before prescribing drug medication [94]. The main and most widely used nutraceuticals, in addition to the aforementioned dietary fibers, are phytosterols and stanols, which reduce the intestinal absorption of exogenous cholesterol and compete with cholesterol in the formation of soluble micelles; probiotics, which improve the excretion of cholesterol in faeces; policosanols, which interfere with the endogenous cholesterol synthesis; soy and lupine proteins, which down-regulate the expression of sterol regulatory element-binding protein (SREBP)-1, reducing the secretion of lipoproteins by the liver and regulating the expression of SREBP-2, increasing cholesterol clearance from the bloodstream; and omega-3 fatty acids, which reduce the hepatic synthesis of VLDL and impair the de novo synthesis of triglycerides.

Another nutraceutical worth mentioning is red yeast rice (RYR), which essentially provides monacolin K, a substance similar to lovastatin, which inhibits 3-hydroxy-3-methyl-glutaryl-coenzyme A (HMG-CoA) reductase; although effective in improving the blood lipid profile, due to its similarity to lovastatin, RYR is considered a drug in some regions of the world, such as in Europe, and when prescribed as a supplement, its monacolin K content must be <3 mg/day [94,95]. Additionally, RYR has not been extensively studied in the pediatric population, so individuals under 18 years of age treated with RYR should be closely monitored both clinically and biochemically.

Some medical devices derived from polysaccharide complexes (Policaptil Gel Retard^®^ and Neopolicaptil Gel Retard^®^) can be used. They appear to have positive effects on many components of MS, including excess weight and various aspects of dyslipidemia [96]. Their effects are multifaceted. From a mechanical point of view, they can sequester a portion of the lipids and sugars consumed in the diet; metabolically, in animal models [97] and subsequently confirmed in humans [96], they appear to modulate lipid homeostasis, metabolism, and storage in the liver, as well as upregulate a large number of genes involved in glucose homeostasis. If the approaches outlined so far do not have any effect, the next step to consider would be the possible use of actual drugs, such as ezetimibe [88] and statins [98]. However, these aspects are beyond the scope of this review.

### 3.3. Obesity, Insulin Resistance, and Type 2 Diabetes Mellitus (T2DM)

It has been shown that children and adolescents with obesity often have higher levels of insulin resistance (expressed as the Homeostasis Model Assessment, HOMA-index) compared with their peers with normal weight [99]. It is known that in this age range, hypertension and insulin resistance are related to one another [100]. Additionally, the HOMA index is an independent predictor of the likelihood of having MUO, and about 15% of children and adolescents with MHO have elevated HOMA-index values [25]. The mechanisms by which insulin resistance promotes the development of hypertension are numerous, ranging from endothelial dysfunction to increased sympathetic activity, which leads to vasoconstriction, the activation of the renin–angiotensin–aldosterone system, and increased tubular sodium reabsorption [101]. Furthermore, in a pediatric population, it has been demonstrated that the HOMA-index plays an important role in mediating the effect that elevated BMI z-scores have on blood pressure levels [71].

In the past twenty years, the prevalence of juvenile-onset type 2 diabetes mellitus (T2DM) has increased parallel to the rise in the number of youths with overweight and low physical activity levels [102]. Obesity, however, with the associated increase in insulin resistance, is a necessary but not sufficient condition for the development of T2DM. As a matter of fact, many individuals with obesity and evident insulin resistance do not develop diabetes at a young age. Adequate glycemic control will fail only when there is a relative (compared with the level of resistance) insulin insufficiency [103], which can occur in the presence of a combination of predisposing factors such as a diet high in calories, saturated or trans fats, and simple sugars, a low level of physical activity, the concomitance of various genetic variants affecting glucose metabolism, certain ethnicities, maternal diabetes and obesity, or a family history of diabetes [104,105]. Breastfeeding, on the other hand, could have a protective effect [106]. A critical moment for the development of juvenile T2DM is the onset of puberty; indeed, cases of T2DM are extremely rare in prepubertal subjects [107]. The physiological increase in insulin resistance that accompanies pubertal development can destabilize an already precarious glucose balance in predisposed adolescents with obesity [104]. In these cases, the latency time between the development of obesity and the onset of T2DM would be about 10 years [108]. Compared with the adult form and type 1 diabetes, youth-onset T2DM has a worse clinical course and more rapid appearance of microvascular and macrovascular complications; this progression can be partly explained by the frequent coexistence of other risk factors related to obesity [109]. In 2017, the prevalence of youth-onset T2DM in the USA was 0.67 cases per 1000 subjects aged 10 to 19 years, with significant differences among various ethnicities [110]. Furthermore, 30% of adolescents are believed to have prediabetes [111]. There are no comprehensive reports on the subject for Europe, but the sporadic data collected in this continent do not seem as negative. In the UK, the prevalence of T2DM was reported to be 0.038 per 1000 youngsters under 18 years, while the incidence of new cases in Germany, Austria, France, and Sweden is only a small fraction of the incidence of type 1 diabetes observed in the same age group [112]. Moreover, these European cases often involved adolescent immigrants. Current migration flows could potentially increase the frequency of youth-onset T2DM in Europe as well. To our knowledge, European data on the prevalence of youth prediabetes are not available, probably because this condition is not always carefully searched for.

Both for the prevention and treatment of youth-onset T2DM and prediabetes, a diet and behavioural approach is fundamental [113]. The nutritional approaches already mentioned to fight other diseases that may cause CVRF (Mediterranean diet, Nordic Diet, DASH, traffic light diet [89]) are also valid for the prevention and treatment of T2DM, as their characteristics fit within what is considered a “healthy diet.” What seems really important is to move away from the Western diet, which, due to its unhealthy nutritional characteristics, is among the most important predisposing factors for all CVRF [50]. In pediatric patients with any degree of obesity, and with one or more other characteristics of the MS, dietary measures can be accompanied by an indication for the prolonged use of one of the medical devices already described, such as Policaptil Gel Retard^®^ or Neopolicaptil Gel Retard^®^, often capable, as already mentioned, of modulating glucose metabolism [96,97,114].

### 3.4. Obesity and Hyperuricemia

In adults, increased uricemia values are associated with a parallel increase in all-cause mortality [115]. In pediatric age, uric acid values are related to an increase in blood pressure [72,116,117], which could be due to a complex interaction between uric acid and insulin resistance [118]. Moreover, dietary and behavioural intervention is less effective in lowering blood pressure values in children with higher baseline uric acid levels [119]. In adults, an increased risk of obesity has been demonstrated in individuals with elevated uricemia levels [120]. Children with obesity also tend to have higher uric acid levels, which can be reduced through weight reduction [121]. Foods that are rich in purines can induce increased uric acid, though these foods are rarely consumed in an excessive manner by children. Simple sugars and particularly fructose, however, even if not containing purines, strongly stimulate uric acid production through ATP degradation [122,123]. In the presence of hyperuricemia, the consumption of simple sugars, particularly through sugary drinks, should be strongly limited [70].

### 3.5. Obesity and Obstructive Sleep Apnea Syndrome

For the physiological development of children, the duration of sleep and its quality are crucial. During sleep, the respiratory system can go through a series of disorders, termed sleep-disordered breathing, of which the most common is obstructive sleep apnea syndrome (OSAS), which is the presence of obstructive apneas or hypopneas caused by recurrent upper airway obstruction [124,125]. Repeated episodes of total or partial upper airway obstruction can lead to a reduction in oxygen saturation throughout the night, sometimes associated with hypercapnia. The prevalence of OSAS in the pediatric age group is about 2 percent, and this syndrome, if not diagnosed and treated, can cause a number of health problems for children including cardiovascular alterations. These are of particular clinical interest because they can impact not only cardiovascular health in childhood but also lead to cardiovascular disease in adult life. In children suffering from OSAS, an increased prevalence of hypertension and left ventricular remodeling, as well as increased oxidative stress, endothelial dysfunction, and systemic inflammation, have been described [126].

Obesity is recognized as an important risk factor for OSAS. While enlarged tonsils and adenoids are the main cause found in normal-weight children, multiple mechanisms may be involved in the onset of OSAS in individuals with obesity. Fat deposition in the subcutaneous tissue surrounding the airways in the cervical region may reduce their caliber, and subcutaneous thoracic and abdominal fat may reduce thoracic distensibility and increase intra-abdominal pressure causing alterations in normal breathing [127,128]. In children with obesity, the phenomenon of OSAS is often associated with diurnal hypoventilation that adds to chronic hypoventilation during sleep [129].

Epidemiological studies show that obesity is associated with an increased risk of OSAS in children and adolescents. However, the exact prevalence of OSAS in the presence of significant excess weight is not well defined and varies in the different studies [129,130,131].

In a child with obesity and a diagnosis of OSAS, weight loss should be the first therapeutic goal and is associated with significant reduction in the number of sleep apneas [132]. Tonsillectomy and/or adenoidectomy may be indicated in cases of hypertrophy, although the removal of tonsils and adenoids may lead to further weight gain. In more severe cases, noninvasive positive pressure nocturnal ventilation can be used in individuals in whom body weight reduction cannot be achieved, and has been shown to be effective in improving gas exchange [124,133].

## 4. Practical Advice for Treating Overweight with the Aim of Cardiovascular Prevention

From the above, it is clear that childhood obesity will lead to serious public health issues, both human and economic [134], especially regarding cardiovascular diseases.

To avoid these risks, it is necessary to plan both generalized preventive interventions aimed at all children to increase their physical activity and improve their diets, and individualized treatments for those already overweight. This way, overweight children could feel more psychologically accepted and not experience individualized intervention as something that makes them feel different from others. Each child should have their cardiovascular risk profile defined based on family and personal history and clinical data, such as the quantification of overweight and the possible presence of other risk factors.

A pediatrician’s commitment to identifying cardiovascular risk factors can be an opportunity to improve family adherence to the treatment of their child’s obesity. Indeed, the finding of a cardiovascular risk factor associated with obesity, which is communicated to parents in a correct and authoritative but not alarming manner, can increase family involvement in treating obesity. The physician’s search for obesity-associated pathologies, even if not found, can make parents reflect on the health risk due to overweight, which is too often still considered a transitory and non-dangerous condition in children. Intervention in children with obesity associated with cardiovascular risk factors should involve support from a secondary care center for management and subsequent follow-up. All steps must be shared with the family pediatrician.

### 4.1. Management by the Secondary Care Center

Conduct a thorough family history to identify issues that may recur in the family (e.g., T2DM, dyslipidaemias, hypertension, etc.). Overweight and smoking habits in the family should be considered carefully, and all efforts should be made to eliminate them. It is particularly useful to have firsthand information about the grandparents’ health, as parents themselves are sometimes too young to have manifested certain diseases. In the personal history, gestational age, birth weight, breastfeeding, weaning method and its acceptance, sleep hours, time dedicated to physical activity, and time spent in sedentary activities are of particular relevance. Significant attention must be given to dietary history, including the frequency of intake of different food groups, and preferences and aversions to certain foods. This investigation allows the identification of important and repeated dietary errors, but also helps in providing dietary plans that, as much as possible, respond to the child’s tastes and habits.Perform anthropometric measurements (weight, height, arm circumference, waist circumference, hip circumference, triceps, and subscapular skinfold thickness) and measure blood pressure. The obtained data must be interpreted based on specific pediatric nomograms, differentiated by sex, age, and height.Conduct blood and instrumental tests. A reevaluation of the child’s blood and instrumental tests should be repeated every 12–18 months.

The results of the investigations are explained to the parents and older children, and a dietary and behavioural plan is proposed. Considering the objective conditions and the family’s availability, solutions are recommended to increase physical activity, mainly in the form of sports for older children and active play for younger ones. A personalized dietary plan, which takes into consideration school commitments and schedules, the availability of a caregiver to prepare meals, family habits, and other potential difficulties, is proposed as a weekly plan according to the child’s and family’s requests. It should be explained that it is not mandatory to apply dietary plans rigidly but, once tried and learned, these models should be considered examples that can be further adapted to the child’s tastes and family habits, within limits that do not undermine the intervention. A nutrition expert presents the dietary plans and takes responsibility for responding to all doubts and questions, even via email at later times.

d.Propose to the child and the family a personalized dietary plan based on the principles described below. Different stages of development, childhood, puberty, and adolescence involve significant changes in energy requirements. Therefore, growing children require a significant amount of energy to support the development of tissues and organs, and all proposed dietary plans must take this into account The personalized approach that takes into consideration specific factors of each individual allows for the development of dietary plans that optimally support growth and health. An important aspect of creating balanced dietary plans that meet the children’s specific nutritional needs is evaluating their energy requirements based on ideal weight determined by growth curves, which provide percentiles based on statistical data collected from reference populations. A child’s ideal weight is often expressed in terms of weight percentile for age, weight for height, or body mass index (BMI) for age [134]. The evaluation of energy requirements should be performed using predictive equations based on age, weight, height, and sex. The Harris–Benedict equation is the most frequently used formula for this need [135]. This formula divides energy demand into two main components: energy expenditure at rest (Resting Energy Expenditure, REE) and related to physical activity (Total Energy Expenditure, TEE). The REE reflects the energy needed to maintain vital functions at rest, while the TEE accounts for the calories consumed during physical activity. Another mathematical approach for calculating energy requirements is the use of the factorial method, which is a well-established practice for assessing the caloric intake necessary to maintain the body’s metabolic functions. The Schofield formula [136], one of the most commonly used variants of this approach, is based on the assumption that energy needs are directly proportional to an individual’s body weight. It involves the use of a multiplicative coefficient that varies according to the individual’s level of physical activity, allowing for a more precise estimate of the calories needed to support metabolic and physical activity [137]. This method provides a useful framework for dietary planning and optimizing caloric intake under various physiological and pathological conditions. In pediatric nutrition, adopting normocaloric dietary schemes is fundamental as it contributes to the healthy physical and cognitive development of children. Normocaloric dietary schemes are designed to provide the right amount of energy needed to support metabolic activities and growth during different developmental stages. Ensuring adequate caloric intake is essential to support the formation of new tissues, bone mineralization, and proper neuro-cognitive development. Moreover, during critical developmental phases such as childhood and adolescence, adequate nutrient intake is crucial to prevent nutritional deficiencies and promote harmonious growth. Normocaloric dietary schemes must be balanced and tailored to the specific needs of each child, considering factors such as age, sex, physical activity, and any existing medical conditions. Designing an optimal dietary scheme for children requires careful consideration of established guidelines for the intake of proteins, lipids, and carbohydrates to ensure healthy and balanced development. General recommendations emphasize the importance of adequate protein intake, typically ranging from 10 to 15% of total calories, accounting for the child’s energy needs and body weight. Proteins should come from nutrient-rich sources such as fish, lean meats, dairy, and legumes. Regarding lipids, guidelines usually recommend that 25–35% of total calories come from fats, with particular attention to essential fatty acids and limiting saturated and trans fats. For carbohydrates, the recommendation is to provide about 45–50% of total calories from this source, favoring complex carbohydrates like fruits, vegetables, and whole grains, while limiting added sugars [138]. Another nutrient to consider when developing a dietary scheme is fiber. The recommended amount of fiber varies according to the child’s age and daily caloric needs. Generally, a gradual increase in fiber intake with age is advised to match increasing nutritional needs. Recommendations suggest that children aged 1 to 3 years should consume about 19 g of fiber per day, while for those aged 4 to 8 years, intake should increase to about 25 g per day. Fiber, mainly derived from fruits, vegetables, legumes, and whole grains, not only promotes intestinal regularity but also helps control blood sugar levels and maintain a healthy body weight [139]. In designing a dietary scheme for children, it is crucial to balance the overall intake of all nutrients to guarantee an adequate integration of vitamins and micronutrients. Vitamins, essential for several biological processes, play key roles in growth, metabolism, and maintaining a robust immune system. Micronutrients, such as minerals and trace elements, are also fundamental for bone health, tissue formation, and the regulation of metabolic reactions. The appropriate intake of vitamins like A, C, D, and B group vitamins, along with minerals such as iron, calcium, and zinc, is essential to prevent nutritional deficiencies that could negatively impact the growth and general well-being of children [138,139,140]. A balanced dietary scheme that includes a variety of foods from all nutritional categories provides a comprehensive approach to ensuring the optimal intake of essential vitamins and micronutrients. An awareness and implementation of these principles in general dietary planning for children is crucial for promoting optimal development and supporting long-term health. Nutritional counseling for families of children at cardiovascular risk is a key element in managing this pediatric population. This process involves active collaboration among health professionals, including nutritionists, pediatricians, and other specialists, with the aim to provide targeted and personalized support. Nutritional education for both children and their families is essential to achieve understanding and the implementation of a specific diet. It is fundamental to engage families in the process of changing dietary habits in order to ensure that dietary recommendations are sustainable in the home context. Teaching children and their families to recognize appropriate portion sizes and to make conscious food choices is crucial for the success of any dietary strategy for the prevention and/or treatment of cardiovascular diseases. Equally important is to provide detailed guidance on the choice of food sources and information on the importance of the quantity and quality of macronutrients. Additionally, nutritional counseling should inform families about the importance of limiting sodium intake and controlling portions. Nutritional counseling should also actively involve parents in promoting an active lifestyle by incorporating physical exercise into the children’s daily routine. Food education, personalized consultation, and regular monitoring are key elements of nutritional counseling, aiming to instill healthy eating habits that can positively influence children’s long-term cardiovascular health.

### 4.2. Control of Blood Pressure Values

Increase the intake of fruits and vegetables for their high contents of potassium, fiber, and antioxidants. Potassium and magnesium contribute to the regulation of blood pressure by improving vascular function and reducing peripheral resistance. The high fiber content in fruits and vegetables may also reduce insulin resistance and increase the feeling of satiety.Reduce sodium intake. Limit foods with high sodium contents, such as processed foods and fast food. Reducing sodium intake has shown direct effects on blood pressure by decreasing water retention and vascular sensitivity. The amount of added salt during daily food preparation should not exceed 5 g (about 2 g of sodium), and even less for younger children. The consumption of foods with much salt should be limited (cured meats, aged cheeses, processed foods, ready meals, foods preserved in brine or salt, and foods with evident added salt such as French fries and popcorn).Consume low-fat dairy products, as they provide calcium without an excess of saturated fats. Additionally, the bioactive peptides present in dairy products may have a vasodilating effect.Reduce the intake of simple sugars. Added sugars, often consumed by children, are present in sugary drinks, sweets, and highly processed foods. Reducing the intake of simple sugars not only promotes the maintenance of a healthy body weight but can also reduce vascular resistance.

### 4.3. Control of Dyslipidaemia

Limit the intake of saturated fats, mainly found in foods of animal origin and in processed food products (red meat, high-fat dairy products, baked goods, and fried foods). Promoting a diet low in saturated fats is essential for lowering LDL cholesterol levels in children with dyslipidaemia. Additionally, these saturated fats can negatively affect lipid metabolism by increasing the production of cholesterol in the liver and reducing the body’s ability to eliminate excess cholesterol.Limit dietary cholesterol intake to no more than 100 mg per 1000 kcal of energy intake with the diet. This goal is easily achievable by including 2–3 “vegetarian” meals per week.Prefer the intake of foods containing unsaturated fats by replacing saturated fats with unsaturated fats. Sources of unsaturated fats, such as fish, olive oil, avocado, nuts, and seeds, can be encouraged in the diet to promote a more favorable lipid profile. Increasing the intake of foods with unsaturated fats is also recommended as they represent a source of omega-3, particularly found in fatty fishlike salmon, trout, and tuna, and in olive oil. Long-chain fatty acids from the omega-3 and omega-6 series can help reduce triglyceride levels.Increase the intake of soluble fiber found in fruits, vegetables, legumes, and grains. These fibers can bind to cholesterol, reducing its intestinal absorption and thus contributing to the reduction of total and LDL cholesterol levels. Since soluble fibers are abundant in a wide range of foods, it is relatively simple to incorporate them into children’s daily diets. The following foods are all excellent source of soluble fiber: fruits like apples, pears, citrus fruits, and bananas; vegetables such as carrots, broccoli, and zucchini; legumes like beans, lentils, and peas; and oats and healthy snacks like fresh fruit or nuts. Incorporating a variety of these foods into the diet can ensure an adequate intake of soluble fiber.

### 4.4. Control of Hyperinsulinemia

Include complex carbohydrates in the diet, such as whole grains, fruits, and vegetables, in order to prevent glycemic spikes and provide energy in a gradual manner.Distribute the intake of carbohydrates throughout the day, both in quantitative and in qualitative terms, to maintain stable blood sugar levels.Limit the intake of added sugars to improve insulin sensitivity and reduce triglyceride and uric acid levels. Completely eliminate sugary beverages.

### 4.5. Obstructive Sleep Apnea Syndrome

As mentioned above, weight reduction is the main measure for reducing and eliminating obstructive sleep apnea associated with obesity.

## 5. Follow Up

Conduct quarterly check-up visits to evaluate the progress of BMI z-score and blood pressure together with the child and parents. Discuss with the pediatrician and nutritionist any difficulties encountered in following the program, specific requests, or additional modifications to the dietary regimen that are needed.

## 6. Conclusions

Pediatric obesity is a high-prevalence condition that leads to serious health damage in the short, medium, and long term and is very difficult to treat. Preventive action to correct the obesogenic environment in which our children grow up can be an effective strategy but only if implemented very early. Parents of children with obesity often do not perceive the problem or underestimate it, believing the child’s excess weight to be “normal” or thinking that the excess weight will resolve itself over time and will not pose a health issue in the future [15]. The reality is quite different; although a small portion of children with obesity will normalize their weight over time without any specific intervention, the majority of overweight children will become adults with obesity. This is even more true the more severe the excess weight is, and when it is present in late childhood or adolescence [141]. It is common for children and adolescents with obesity to present with one or more cardiovascular risk factors, such as hypertension, dyslipidaemia, hyperinsulinemia, carbohydrate metabolism disorders, or hyperuricemia [142]. These possible conditions cause much concern for the families, as cardiovascular risk factors are perceived as diseases, whereas the presence of obesity does not have this effect. Even in a child who does not yet present these alterations, the very fact that the pediatrician is looking for them can make parents reflect on the real impact that childhood obesity has on health. In this way, excess weight is not treated in itself but as a possible cause of diseases. It is important to convey the presence of conditions related to excess weight with competence and authority but without creating unnecessary apprehension in parents, which could lead them to deny reality and refuse any intervention that is proposed. For these reasons, it is always necessary to emphasize that improvement is possible, that there is time to achieve it, that it is not necessary to reach normal weight in order to reduce cardiovascular risk factors but that even partial improvements can be important for the children’s future health. The most effective interventions in children with obesity are those implemented by multidisciplinary teams. Our experience suggests that these teams should be composed of pediatricians, dieticians and/or nutritionists, and also cardiologists, for the detection of any early cardiovascular risk factors. Therefore, it is appropriate for the child/adolescent with obesity to undergo an initial evaluation through laboratory and instrumental tests aimed at searching for these risk factors, which, if present, place the child with obesity in a higher risk range. The team of specialists should also have easy access to psychological counseling, as obesity can often be an effect or cause of a psychological disorder, which can be reflected in the child/adolescent’s behaviour, often worsening overweight. A physical activity expert should also join the multidisciplinary team. The overweight treatment pathway is very difficult for children/adolescents and their families. This concept needs to be very much in the minds of clinicians dealing with this issue and it is essential to promote empathetic collaboration between clinician and patients.

## Data Availability

Not applicable.

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
