# Peer review of "Non-Pharmacological Treatment for Cardiovascular Risk Prevention in Children and Adolescents with Obesity"

_nutrients, 2024, doi:10.3390/nu16152497_

Round 1

Reviewer 1 Report

Comments and Suggestions for Authors

The manuscript entitled Non-pharmacological treatment for cardiovascular risk prevention in obese children and adolescents is a narrative review. The authors realized an update on non-pharmacological interventions to be implemented for cardiovascular prevention in obese children and adolescents and their effectiveness.

Nowadays, obesity in children and adolescents is a major issue in clinical practice. Therefore, this manuscript could be important for readers. It is well written, well organized.

Something could create confusion in this manuscript: Metabolically Healthy Obesity. Please, confine this term, especially because there are pediatric societies, which does not recognize this term.

I suggest to introduce in the manuscript the term paradoxical obesity and to make the differentiation between these two terms.   

I recommend introducing in chapter 2 (page 2) a table with the main publications about obesity as cardiovascular risk factor.

I recommend introduce some data about obesity and sleep apnea in this type of population, because this association means, especially, arrhythmias.

There are some English language errors (For ex.: Page 7 line 361).

Author Response

The manuscript entitled Non-pharmacological treatment for cardiovascular risk prevention in obese children and adolescents is a narrative review. The authors realized an update on non-pharmacological interventions to be implemented for cardiovascular prevention in obese children and adolescents and their effectiveness.

Nowadays, obesity in children and adolescents is a major issue in clinical practice. Therefore, this manuscript could be important for readers. It is well written, well organized.

Comment 1: Something could create confusion in this manuscript: Metabolically Healthy Obesity. Please, confine this term, especially because there are pediatric societies, which does not recognize this term.

Response 1: Thank you for your valuable remark. It actually reflects perfectly our point of view. We tried to make it clear throughout the whole manuscript that we do not believe in this definition, and we rather consider the Metabolically Healthy Obesity as ‘still healthy’ obesity. We apologize if we could not make the point clear enough. To avoid confusion, we have added a paragraph and two new references to better clarify this concept.

Comment 2: I suggest to introduce in the manuscript the term paradoxical obesity and to make the differentiation between these two terms.   

Response 2: Thank you for your suggestion. However, we are a bit confused about it. The paradox of obesity is a concept well-recognized in the field of adult obesity, but we could not find a similar concept in the literature about childhood and adolescent obesity. We would be grateful if you could clarify what you mean by this point. 

Comment 3: I recommend introducing in chapter 2 (page 2) a table with the main publications about obesity as cardiovascular risk factor.

Response 3: We understand the reviewer's suggestion, but we think that the studies on this topic are so numerous that it is impossible to make a Summary Table, which would risk on the one hand being incomplete and on the other hand weighing down the manuscript too much. We think it is best for readers to refer to the extensive bibliography given.

Comment 4: I recommend introduce some data about obesity and sleep apnea in this type of population, because this association means, especially, arrhythmias.

Response 4: We thank the reviewer for this important suggestion. Obstructive sleep apnea syndrome is certainly an important condition associated with obesity and should be mentioned in a review dealing with pediatric obesity and cardiovascular disease. We have therefore added a chapter on OSAS and included a number of references regarding this topic.

Comment 5: There are some English language errors (For ex.: Page 7 line 361).

Response 5: Thank you, errors have been corrected.

Reviewer 2 Report

Comments and Suggestions for Authors

The work provides an excellent review of the problem of obesity and overweight in adolescents.

This is a broad description with a high number of bibliographical references.

The information that would be convenient to provide in the conclusions is how to resolve some of the questions that the manuscript raises, not by giving a solution but by providing future strategies to consider.

For example, the approach to psychological support in treating pediatric obesity.

Lifestyle and nutrition changes to treat obesity in children and adolescents, firm proposals for protocols to follow

Table 1, although important, does not contribute anything, unless similar or more sociodemographic tables are made with the rest of the parameters analyzed.

It is suggested that the table be reformulated, eliminated, or proposed to create others with the other data.

Author Response

The work provides an excellent review of the problem of obesity and overweight in adolescents.

Thank you for your appreciation.

This is a broad description with a high number of bibliographical references.

Thank you, we are glad to read that you appreciate our efforts in gathering a robust bibliography.

Comment 1: The information that would be convenient to provide in the conclusions is how to resolve some of the questions that the manuscript raises, not by giving a solution but by providing future strategies to consider. For example, the approach to psychological support in treating pediatric obesity. Lifestyle and nutrition changes to treat obesity in children and adolescents, firm proposals for protocols to follow

Response 1: Thank you for your valuable comment. Yes, you are totally right, and we added some sentences in the conclusions, to address your suggestion. We believe that the strategies to be addressed should include not only the psychological support, but also a better evaluation of the potential risks, and also a lifestyle modification strategy driven and sustained by the managing team. Some sentences have been added to the conclusions to emphasize these concepts more.

Comment 2: Table 1, although important, does not contribute anything, unless similar or more sociodemographic tables are made with the rest of the parameters analyzed. It is suggested that the table be reformulated, eliminated, or proposed to create others with the other data.

Response 2: We agree with the reviewer and decided to remove the table, the contents of which are already described in the text.
